# Chronic replication stress-mediated genomic instability disrupts placenta development in mice

**Mumingjiang Munisha[1], Rui Huang [1], Jordan Khan [2], John C. Schimenti [1]\***

**1** Cornell University, College of Veterinary Medicine, Department of Biomedical Sciences, Ithaca, New York, United States of America, **2** Cornell University, Nancy E and Peter C Meinig School of Biomedical Engineering, Ithaca, New York, United States of America

☯ These co-authors contributed equally to the manuscript.
\* jcs92@cornell.edu

## Abstract

Abnormal placentation drives many pregnancy-related pathologies and poor fetal outcomes, but the underlying molecular causes are understudied. Here, we show that persistent replication stress due to mutations in the MCM2–7 replicative helicase disrupts placentation and reduces embryo viability in mice. MCM-deficient embryos exhibited normal morphology, but their placentae had a drastically diminished junctional zone (JZ). Whereas cell proliferation in the labyrinth zone (LZ) remained unaffected, it was reduced in the JZ during development. MCM2–7 deficient trophoblast stem cells (TSCs) failed to maintain stemness, suggesting that replication stress affects the initial trophoblast progenitor pool in a manner that preferentially impacts the developing JZ. In contrast, pluripotency of mouse embryonic stem cells (ESCs) with MCM2–7 deficiency were not affected. Developing female mice deficient for FANCM, a protein involved in replication-associated DNA repair, also had placentae with a diminished JZ. These findings indicate that replication stress-induced genomic instability compromises embryo outcomes by impairing placentation.

## Author summary

Proper placental development is essential for the health of both the mother and the fetus during pregnancy. In this study, we show that genetic defects in the DNA replication machinery - resulting in genomic instability - can disrupt placental formation at very early stages of development. Using mouse models with ongoing DNA replication stress, we found that early placental progenitor cells, called trophoblast stem cells, lose their ability to self-renew. This leads to a reduced pool of placental stem cells and a shortage of key types of differentiated trophoblasts. Consequently, placentas in mutant fetuses are smaller and poorly developed, which negatively affects embryo growth, survival, and the health of survivors. Notably, female fetuses are affected more severely than males.

**Data availability statement:** RNA-Seq data was deposited to the Gene Expression Omnibus with the accession number GSE290784.

**Funding:** This work was supported by the National Institutes of Health (5R01HD107910 to JCS), the Druckenmiller Postdoc Fellowship from the New York Stem Cell Foundation (to MM) and the Lalor Foundation (to MM). The funders had no role in study design, data collection and analysis, decision to publish, or preparation of the manuscript. These recipients salaries were partially funded by these awards.

**Competing interests:** The authors have declared that no competing interests exist.

Although the placenta is a temporary organ that normally displays a relatively high level of genetic anomalies, our findings in mice suggest that protecting the genome from excessive damage during the earliest stages of development is critical for building a healthy placenta that can support a successful pregnancy. These results underscore the importance of further studying how genome maintenance pathways contribute to placental development and how their disruption may lead to fetal loss or intrauterine growth restriction.

## Introduction

Early embryonic development is characterized by rapid cellular proliferation and differentiation. In such a period, it is imperative for cells, especially stem cells that engender various components of the developing embryo, to maintain genomic integrity to promote proper formation, propagation and maintenance of cellular identity [1,2]. Genome maintenance requires precise coordination of various cellular processes including those involved in DNA replication and repair [3,4]. Disruptions to these processes can lead to genomic instability (GIN), a cellular state characterized by elevated DNA damage such as double-strand breaks (DSBs), cell cycle delay/arrest and mutation accumulation [4].

Most of our knowledge of basic genome maintenance mechanisms has been derived from studies on cancer cell lines. The impact of GIN upon organismal development has been focused primarily on the embryo proper. Although extraembryonic placental cells share several characteristics with cancer cells such as invasiveness and high levels of copy number variation [5], the mechanisms underlying genome maintenance in placental cells are relatively understudied. For example, mouse trophoblast giant cells (TGCs) are highly polyploid and selectively amplify regions containing key placental genes, causing extensive copy number variation throughout the genome [6–8]. Differentiated trophoblasts are resistant to GIN as they do not activate DNA damage checkpoints in response to genotoxic stress [9–12]. This is understandable given the transient nature of the placenta. Similarly, studies using term human placental tissues have revealed that high mutation rates, genome amplification and senescence are common features of human trophoblast cells [13–15]. Mutations in genes involved in genome maintenance can lead to diseases [3]. Though such mutations are rare, mammalian embryos and placentae face unique challenges during normal development that can impose GIN [1]. While excessive GIN can result in adverse embryo outcomes by negatively affecting placental function via inducing senescence and inflammation [16,17], the extent to which GIN is tolerated at different stages of placenta development needs further investigation.

Using a unique mouse model with high intrinsic GIN, known as *Chaos3 (C3)*, we showed that deficiencies in the DNA replicative helicase can result in sex- and parent-of-origin dependent embryonic semi-lethality [17]. The *C3* allele is a hypomorphic missense point mutation in the Minichromosome maintenance complex 4 (*Mcm4)* gene which encodes a protein that is part of the core catalytic unit of the

CMG (CDC45, MCM2–7, and GINS complex) replicative helicase [18]. MCMs are loaded as heterohexamers of MCM2–7 onto DNA replication origins (sites of replication initiation) during the G1 phase of the cell cycle, a process referred to as origin "licensing" [19,20]. Only a portion of licensed origins initiate bidirectional replication during S-phase, leaving the rest to serve as dormant origins (DO). These DOs are crucial when cells experience replication stress (RS), enabling rescue of chromosomal regions left unreplicated from stalled or collapsed replication forks [19,20]. The amino acid change in MCM4$^{C3}$ disrupts its interaction with MCM6, destabilizing the MCM2–7 complex. This leads to persistent RS and activation of p53-dependent upregulation of miRNAs that target *Mcm* mRNAs, reducing DOs, increasing hypersensitivity to DNA damaging agents, and dramatically increasing cancer susceptibility [18,21,22].

*Mcm4*$^{C3/C3}$ mice are viable and fertile, and exhibit elevated micronuclei (a nuclear membrane-bound cytosolic DNA) in erythrocytes, a hallmark of persistent RS and GIN [18]. Genetic depletion of another MCM in the *Mcm4*$^{C3/C3}$ mice (for example with the genotype *Mcm4*$^{C3/C3}$ *Mcm2*$^{Gt/+}$; "*Gt*" is a gene trap null allele) resulted in embryonic semi-lethality in which 40% of males survive to birth compared to only 15% of females [17,22]. The female lethal bias was associated with elevated markers of placental inflammation, and the sex skewing could be eliminated by suppressing inflammation pharmacologically [17]. We hypothesized that the inflammation and female-biased embryonic lethality was caused by activation of innate immune pathways that sense cytosolic nucleic acids. Here, we report that key cytosolic DNA sensors (cGAS-STING; RIG1-MAVs) are not independently responsible. Rather, elevated RS causes defective trophoblast stem cell (TSC) proliferation/maintenance in mutants with high levels of GIN resulting in defective placentation, and inflammation is likely a consequence of this disruption. Overall, these studies indicate that placentation is sensitive to high amounts of RS-induced GIN during early development.

## Results

### Persistent replication stress (RS) leads to a small placenta and intrauterine growth restriction

As mentioned earlier, deficiencies in DNA replicative helicase causes embryonic semi-lethality, preferentially affecting female embryos [17]. Female-biased embryonic lethality was observed in mice bearing the semi-lethal genotype *Mcm4*$^{C3/C3}$ *Mcm2*$^{Gt/+}$ from the sex-skewing mating pairs, where the dam genotype is *Mcm4*$^{C3/C3}$ and the sire genotype is *Mcm4*$^{C3/+}$ *Mcm2*$^{Gt/+}$ (Fig 1A). However, this female bias was abolished in the reciprocal mating where the dam and sire genotypes were switched (Fig 1A). We previously proposed that placental inflammation might be the culprit of female-biased embryonic semi-lethality for the following reasons: 1) RNA-Seq of E13.5 placenta revealed an enrichment of inflammatory signatures; 2) anti-inflammatory (Ibuprofen or testosterone) treatment of pregnant dams rescued the female-bias; and 3) deficiency for the anti-inflammatory gene *Il10rb* was synthetically lethal with *Mcm4*$^{C3/C3}$, but viability could be rescued by ibuprofen treatment of pregnant dams [16]. Given that elevated micronuclei is the hallmark of *Mcm4*$^{C3/C3}$ mice [18], we hypothesized that a cytosolic nucleotide sensing pathway, such as cGAS-STING, was responsible for increased placental inflammation, and that male embryos were preferentially protected by high levels of testosterone around the time of death. To test this hypothesis, we genetically deleted *Tmem173* (encoding STING protein in the cGAS-STING pathway) and separately *Ddx58* (encoding RIG1 protein in the RIG1-MAVS pathway) in the *Mcm4*$^{C3/C3}$ *Mcm2*$^{Gt/+}$ background. Deletion of neither gene rescued female-biased embryonic lethality (S1 Table). Additionally, knocking out *Myd88*, encoding an obligate co-factor for the Toll-Like-Receptor signaling, in the *Mcm4*$^{C3/C3}$ background was synthetically lethal post-wean, not permitting us to generate animals of the desired genotype.

To further investigate the cause of female-biased embryonic lethality, we assessed embryonic and placental growth during mid-gestation in both sex-skewing and reciprocal matings (Fig 1A). At E13.5, *Mcm4*$^{C3/+}$ embryos and placentae from the sex-skewing mating were grossly indistinguishable from WT (produced from WTxWT matings) (Fig 1B). However, *Mcm4*$^{C3/C3}$ *Mcm2*$^{Gt/+}$ placentae and embryos from the sex-skewing mating were significantly smaller than *Mcm4*$^{C3/+}$ littermates and *Mcm4*$^{+/+}$ controls (Fig 1B). Both embryonic and placental weights were significantly reduced in *Mcm4*$^{C3/C3}$ *Mcm2*$^{Gt/+}$ genotypes from both sex-skewing and reciprocal matings (Figs 1C-E and S1A-S1C), but the reductions in

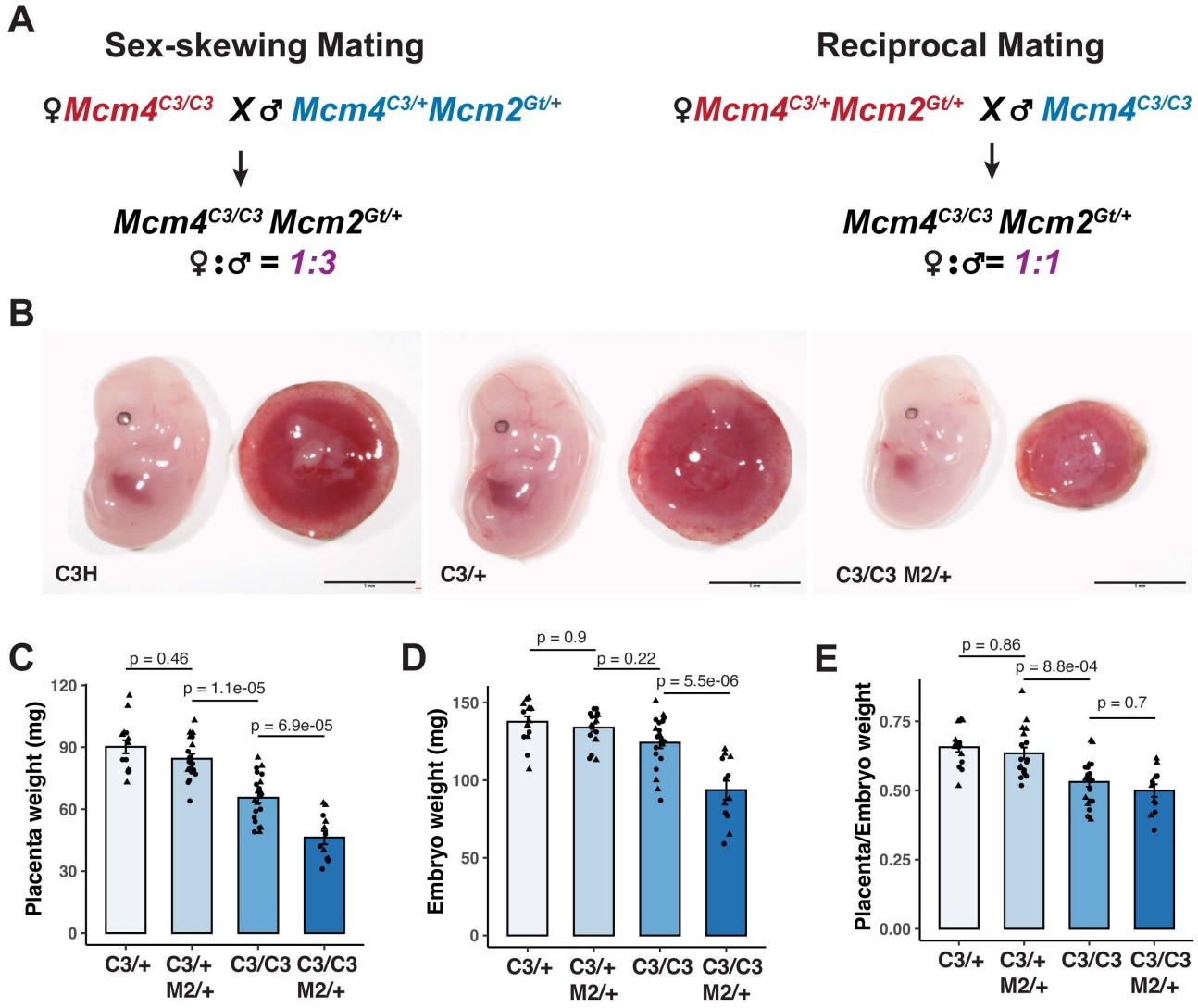

**Fig 1. Semi-lethal genotype mutants have reduced placental and embryonic weight. (A)** Mating scheme for sex-skewing and reciprocal crosses to generate semi-lethal genotypes. **(B)** Representative images of embryos with indicated genotypes. Scale bar: 5mm. **(C-E)** Placental weight, embryonic weight, and placental-to-embryonic weight ratio at E13.5 from sex-skewing matings. C3/+: *Mcm4^C3/+*; C3/C3: *Mcm4^C3/C3*; C3/+M2/+: *Mcm4^C3/+ Mcm2^Gt/+*; C3/C3 M2/+: *Mcm4^C3/C3 Mcm2^Gt/+*. p-values were calculated with one-way ANOVA with Tukey's HSD test. ns: not significant. Error bars: SEM. Each data point represents a single placenta or embryo; females (circles), males (triangles).

females were more severe in sex-skewing matings than in reciprocal matings. Males of the same genotype didn't show parent-of-origin dependent placental weight reduction (S1D Fig). In addition, the placenta-to-embryo weight ratio was significantly reduced in *Mcm4^C3/C3* and *Mcm4^C3/C3 Mcm2^Gt/+* genotypes from both sex-skewing and reciprocal matings at E13.5, suggesting that the growth in placenta was hindered in high GIN genotypes (Figs 1E and S1C).

To investigate whether the effect of GIN on placental development is unique to the *Chaos3* model, we collected placentae and embryos from *Fancm^+/-* intercrosses during mid-gestation and measured their weights. FANCM is involved in DNA replication fork repair. Similar to *Chaos3* mice, FANCM deficiency also presents with increased GIN and female-biased lethality [17,23]. At E13.5, both placental and embryonic weights of *Fancm^-/-* animals were significantly reduced compared

to control littermates (S2A-B Fig). These data suggested that the developing trophoblast lineage is particularly sensitive to persistent replication stress.

**Semi-lethal mutants exhibit placental developmental defects preferentially affecting the junctional zone (JZ)**

To determine whether the reduction in placental weight in semi-lethal genotypes was related to any underlying structural defects, we carried out histological analysis of placentae at E13.5. Periodic Acid Schiff (PAS) staining showed significantly reduced JZ and labyrinth zone (LZ) areas in semi-lethal genotype placentae compared to control littermates from both sex-skewing and reciprocal matings (Figs 2A-C and S3A-D). Within the JZ, PAS-positive glycogen trophoblast cells (GlyTs) were almost absent in $Mcm4^{C3/C3}$ $Mcm2^{Gt/+}$ placentae from sex-skewing matings (Fig 2A). Additionally, spongio-trophoblasts (SpTs) and parietal Trophoblast Giant Cells (p-TGCs, which have distinctly large nuclei compared to diploid cells) within the central regions of the maternal-fetal interface were severely depleted (Fig 2A). On the other hand, some SpTs were present in the $Mcm4^{C3/C3}$ $Mcm2^{Gt/+}$ placentae from reciprocal matings (S3A Fig). Additionally, we found that the reduction in the JZ area was more prominent in females with the semi-lethal genotype compared to males from sex-skewing matings (S3E-G Fig). In contrast, no sex differences in JZ areas were observed in placentae from reciprocal matings (S3E Fig). The JZ:total placental area (JZ + LZ) ratio was significantly reduced in semi-lethal genotypes (Fig 2D) with a more prominent reduction in females from sex-skewing matings (S3G Fig). Female, but not male, $Fancm^{-/-}$ placentae also had a reduced JZ, but the LZ appeared normal (S2C-D Fig).

To determine whether the diminished LZ was due to a decreased number of trophoblasts, we immunostained E13.5 placentae with MCT1 and MCT4 that are markers of Syncytiotrophoblasts I and II (SynTI and SynTII) respectively. The overall MCT1- and MCT4-positive areas were reduced in $Mcm4^{C3/C3}$ $Mcm2^{Gt/+}$ genotypes compared to control littermates (Fig 2E), in agreement with the observed reduction in LZ size. However, the proportions of MCT1 and MCT4 positive areas within the LZ in $Mcm4^{C3/C3}$ $Mcm2^{Gt/+}$ genotypes were not different from the control littermates (Fig 2F-G).

We next performed bulk RNA-Seq of E13.5 placental samples as a means to corroborate the histological observations, exploiting differentially expressed genes identified as being diagnostic for various placental cell types [24]. Additionally, to account for cellularity differences in the mutant placentae, we analyzed a publicly available single-cell RNA-Seq (scRNA-Seq) dataset from E13.5 mouse placenta [25] (S4A-B Fig), then deconvoluted our bulk RNA-Seq results using BayesPrism [26], a Bayesian statistical model for estimating the proportion of cell types using benchmark gene expression profiles (in this case, the aforementioned E13.5 scRNA-seq). Semi-lethal genotype placentae exhibit significantly reduced GlyT cell type fractions, affecting the female $Mcm4^{C3/C3}$ $Mcm2^{Gt/+}$ placentae compared to controls, while other trophoblast cell types showed no significant difference. SpTs showed a reduced trend in mutant females compared to controls (S4D Fig). The proportions of certain trophoblast subtypes, such as p-TGCs versus S-TGCs and SynTI versus SynTII, were difficult to accurately estimate in bulk samples due to their highly similar gene expression profiles (S4C-D Fig). Furthermore, the percentage of proliferating cells in the JZ, but not in the LZ, was significantly reduced in $Mcm4^{C3/C3}$ $Mcm2^{Gt/+}$ placentae as indicated by Ki67 immunostaining, and was negatively correlated with the level of GIN in litters from the sex-skewing mating (Fig 3A-B). Collectively, these data suggest that the JZ trophoblast cells are particularly susceptible to chronic RS mediated by MCM deficiency.

Additional gene set enrichment analysis (GSEA) of the bulk RNA-Seq data reflected disruption of the trophoblast compartment. Some of the placenta development-related Gene Ontology (GO) biological processes showed differences between female and male $Mcm4^{C3/C3}$ $Mcm2^{Gt/+}$ placentae. Labyrinth-related pathways remained the same, whereas JZ-related pathways (such as trophoblast giant cell differentiation and spongiotrophoblast layer development) were under-represented in female mutants compared to males (Fig 3C). Consistent with the histological assessment, expression of genes involved in these underrepresented biological processes were significantly lower in female $Mcm4^{C3/C3}$ $Mcm2^{Gt/+}$ placentae compared to WT (Fig 3D). These differences were less pronounced in male mutants (Fig 3D). Some of the shared genes in these pathways are predominantly expressed in the JZ trophoblast, such as $Ascl2$ [27], $Gjb5$ [28], and $Hand1$ [25], the deficiency of which result in poor JZ development [28–30].

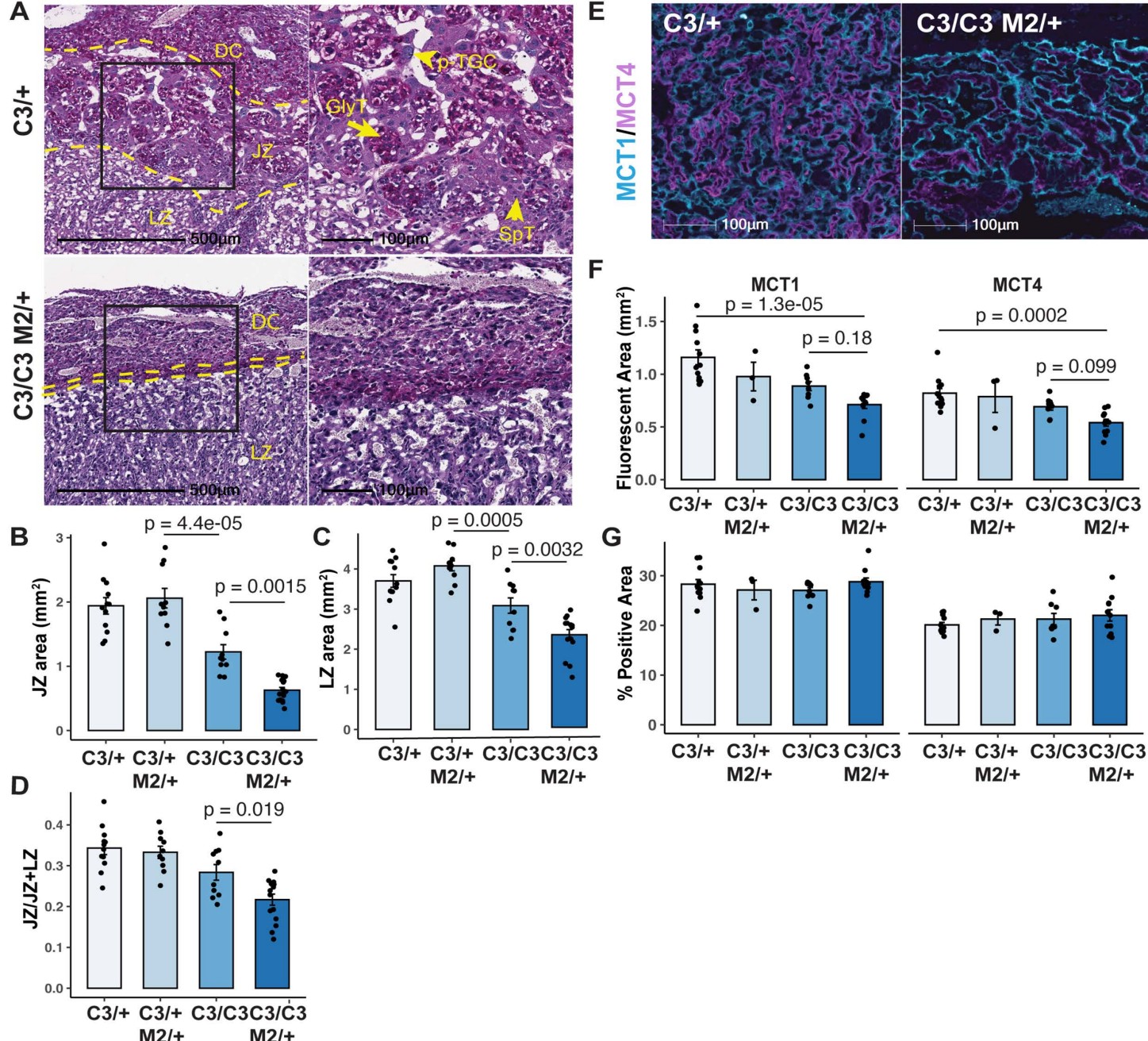

**Fig 2. Semi-lethal genotype placentae have smaller junctional and labyrinth zones. (A)** Periodic Acid-Schiff staining of E13.5 placental sections from indicated genotypes. The middle and left panels show magnified views of the boxed regions in the right panel. **(B-D)** Quantification of junctional zone (JZ) and labyrinth zone (LZ) areas and JZ proportion in placentae from sex-skewing matings at E13.5. **(E)** Representative MCT1 and MCT4 staining of E13.5 placental sections. **(F)** Quantification of MCT1 and MCT4 fluorescent areas, and **(G)** the percentage of positive areas in placental sections of indicated genotypes.C3/+: $Mcm4^{C3/+}$; C3/C3: $Mcm4^{C3/C3}$; C3/+M2/+: $Mcm4^{C3/+}$ $Mcm2^{Gt/+}$; C3/C3 M2/+: $Mcm4^{C3/C3}$ $Mcm2^{Gt/+}$. DC: decidua; JZ: junctional zone; LZ: labyrinth zone; p-TGC: parietal trophoblast giant cells; SpT: spongiotrophoblast; SynT: syncytiotrophoblast; GlyT: glycogen trophoblast. p-values were calculated with one-way ANOVA with Tukey's HSD test. ns: not significant. Error bar: SEM. Each data point represents the average of at least three sections from the same placenta.

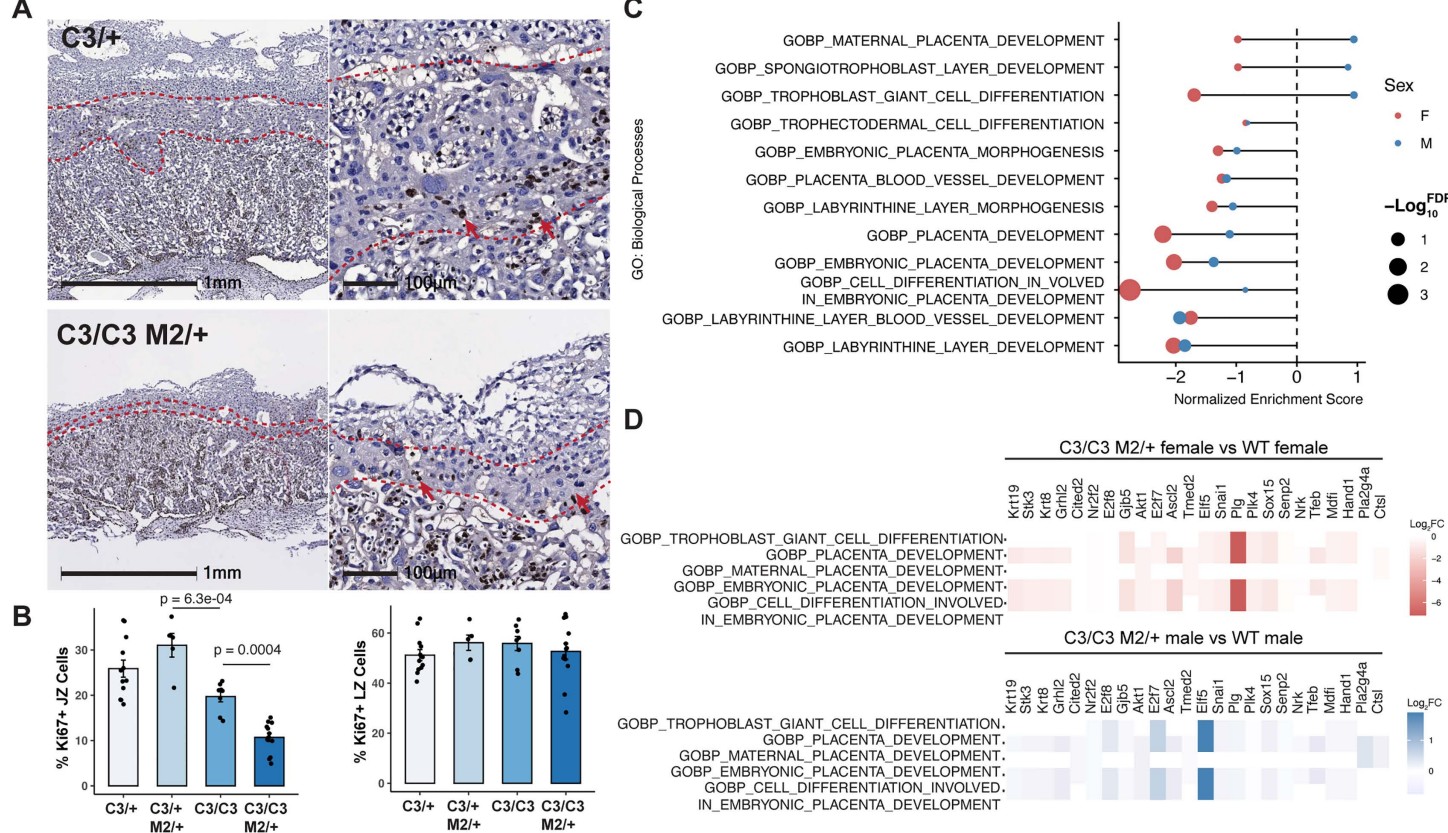

**Fig 3. The JZ in semi-lethal genotype placenta is affected more than the LZ. (A)** Representative Ki67 immunohistochemistry of E13.5 placental sections from indicated genotypes. **(B)** Quantification of Ki67-positive cells in the JZ and LZ from sex-skewing matings. Error bar: SEM. Each data point represents the average of at least three sections from the same placenta. **(C)** Gene Ontology (GO) enrichment analysis of placenta development-related terms. **(D)** Log-transformed fold change of shared placental development genes in females and males. C3/+: $Mcm4^{C3/+}$; C3/C3: $Mcm4^{C3/C3}$; C3/+M2/+: $Mcm4^{C3/+} Mcm2^{Gt/+}$; C3/C3 M2/+: $Mcm4^{C3/C3} Mcm2^{Gt/+}$. p-values were calculated with one-way ANOVA with Tukey's HSD test. ns: not significant.

## Semi-lethal genotypes impair trophoblast stem cell (TSC) establishment and maintenance

At E11.5, we found no evidence of elevated apoptosis (S5A Fig) in the mutant placentae. However, JZ cells were almost absent at this stage, suggesting that defects in the spongiotrophoblast layer occurred earlier during development. At E9.5, TPBPA, a marker of JZ trophoblast, was expressed in a smaller region of $Mcm4^{C3/C3} Mcm2^{Gt/+}$ placentae compared to control littermates, and few apoptotic cells were observed in the spongiotrophoblast layer (S5B-C Fig). These data suggest that defects in $Mcm4^{C3/C3} Mcm2^{Gt/+}$ placentae likely occurred early, either from impaired differentiation ability of trophoblast progenitors or defects in the initial maintenance of the TSC pool.

To test these hypotheses, we performed experiments to derive mutant TSC lines that could be used for growth and differentiation studies. Blastocysts obtained from intercrosses of $Mcm4^{C3/+}$ mice were cultured in conventional TSC derivation media containing FGF4 and heparin on a mouse embryonic fibroblast (MEF) feeder layer [31]. The proportion of $Mcm4^{C3/C3}$ TSCs obtained was much lower than the expected Mendelian ratio (S2 Table), and these lines were difficult to maintain. In an attempt to improve derivation and/or growth efficiency, we next derived TSCs on Matrigel in a chemically defined medium containing growth factors and inhibitors [32,33] (S6A Fig). Again, $Mcm4^{C3/C3}$ lines were

established at sub-Mendelian ratios (S2 Table), but they proliferated more stably under chemically defined conditions. Attempts to derive *Mcm4*^C3/C3^ *Mcm2*^Gt/+^ TSC lines from crosses of *Mcm4*^C3/C3^ females to *Mcm4*^C3/C3^ *Mcm2*^Gt/+^ males were only successful using the defined condition with Matrigel. These findings indicate that high RS is an impediment to TSC growth or maintenance at very early stages, at least *in vitro*. To evaluate the differentiation capacity of mutant TSCs, we initiated spontaneous differentiation by removing all the growth factors and inhibitors from the TSC culture medium. Starting on day 7 (D7) of differentiation, TGCs (PL-1) and SpTs, GlyT (TPBPA) markers can be seen with subsequent loss of the TSC marker EOMES (S6B Fig), indicating that the JZ defect is unlikely a result of defective TSC differentiation.

We next performed EdU pulse-labeling to assess the impact of MCMs deficiency on TSC proliferation. Both *Mcm4*^C3/C3^ and *Mcm4*^C3/C3^ *Mcm2*^Gt/+^ TSCs displayed significantly fewer EdU⁺EOMES⁺ cells than WT, indicating a diminished proliferative capacity (Fig 4A-B). In parallel, cleaved-Caspase 3 staining revealed a substantial increase of apoptotic cells in both *Mcm4*^C3/C3^ and *Mcm4*^C3/C3^ *Mcm2*^Gt/+^ TSCs (Fig 4C-D), coupled with a significantly reduced percentage of EOMES⁺ cells compared to WT TSCs (Fig 4E-F), suggesting a loss of stemness in TSCs with high GIN.

In contrast to the deficiencies in mutant TSC maintenance, both *Mcm4*^C3/C3^ and *Mcm4*^C3/C3^ *Mcm2*^Gt/+^ mouse embryonic stem cell (ESC) lines were readily established from blastocysts and successfully maintained; we observed no differences in the expression of naive pluripotency marker KLF4 among WT, *Mcm4*^C3/C3^ and *Mcm4*^C3/C3^ *Mcm2*^Gt/+^ ESCs (S7A-B Fig). Proliferation rate, assessed by the proportions of EdU + KLF4 + cells, were also comparable between GIN genotypes and WT ESCs (S7C-D Fig). Furthermore, the number of γH2Ax foci and cell death assessed by cleaved Caspase-3 flow cytometry showed no difference across different genotypes (S7E-H Fig). Consistent with the previous findings [34], our data suggest that MCM deficiency does not affect ESCs pluripotency or self-renewal. In summary, we found that intrinsic RS impairs TSC, but not ESC, viability, proliferation and maintenance *in vitro*.

## TSCs with intrinsic RS exhibit activation of the DNA damage response and premature differentiation

To further elucidate the possible molecular mechanisms underlying the observed defects in TSCs with high GIN, we hypothesized that TSCs activate cell cycle checkpoints via the DNA damage response (DDR). As previously demonstrated for mutant placentae [17], both *Mcm4*^C3/C3^ and *Mcm4*^C3/C3^ *Mcm2*^Gt/+^ TSCs contained substantially more γH2AX (a marker of strand breaks, primarily DSBs) than WT cells (Fig 5A-B). In contrast, WT, *Mcm4*^C3/C3^ and *Mcm4*^C3/C3^ *Mcm2*^Gt/+^ ESCs showed similar γH2AX foci per nucleus (S7E-F Fig). This suggests that ESCs are less sensitive than TSCs or do not have a robust DDR in response to RS as a consequence of MCM deficiency. Cultures of TSCs with high levels of GIN were significantly enriched for the G2/M population, suggesting the activation of checkpoint signaling pathways (Fig 5C-D). In addition, both *Mcm4*^C3/C3^ and *Mcm4*^C3/C3^ *Mcm2*^Gt/+^ TSCs displayed a significant increase in the population of cells with >4C DNA content (indicative of polyploidy) relative to WT (Fig 5C-D), suggestive of increased endoreduplication, a process in which repeated rounds of DNA replication occur without mitosis, and which is a remarkable characteristic of mouse TGCs [12].

Given the observed cell cycle disruption in TSCs, we investigated the signaling pathways involved in regulating the G2/M transition. Western blot analysis revealed significantly elevated levels of phosphorylated CHK1 (S345) and phosphorylated p53 (S15) in *Mcm4*^C3/C3^ and *Mcm4*^C3/C3^ *Mcm2*^Gt/+^ TSCs compared to WT (Fig 5A-B). p21 level was increased in *Mcm4*^C3/C3^ *Mcm2*^Gt/+^ TSCs but were not statistically significant (Fig 5A-B). Furthermore, mutant TSCs exhibited a reduction in Cyclin-dependent kinase 1 (CDK1) levels, along with an increased Phospho-CDK1 (Thr14/Tyr15) to total CDK1 ratio, suggesting enhanced CDK1 inhibition (Fig 5E-F). Inhibition of CDK1, which drives the G2/M transition and is a downstream target of p21 (CDKN1A), is required for cells to enter endoreduplication and thereby TGC differentiation [9,35]. Collectively, these findings indicate that persistent RS in TSCs activates the CHK1-p53-p21 axis, leading to CDK1 inhibition and premature differentiation via endoreduplication. Normally, eukaryotic cells have a robust mechanism to ensure

PLOS Genetics

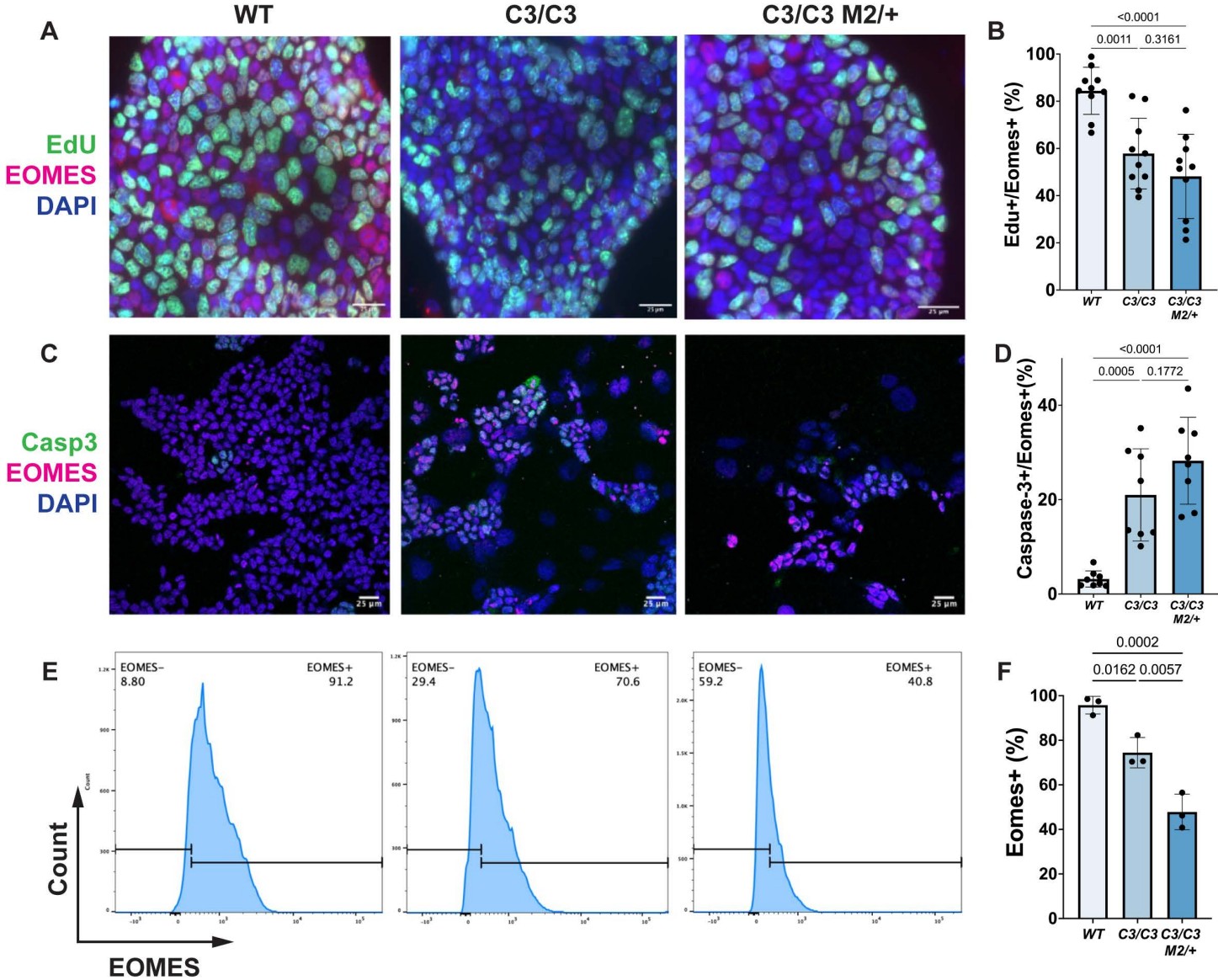

**Fig 4. Semi-lethal genotype trophoblast stem cells (TSCs) exhibit abnormal cellular phenotypes. (A)** EdU pulse labeling of TSCs with immunofluorescence staining for the EOMES. Scale bar: 25 µm. **(B)** Quantification of EOMES and EdU-double positive cells. **(C)** Immunofluorescence staining of TSCs for the Cleaved Caspase-3 and EOMES. **(D)** Quantification of EOMES and Cleaved Caspase-3-double positive cells. **(E)** Flow cytometry analysis of EOMES-positive TSCs. **(F)** Quantification of EOMES-positive cells. C3/+: $Mcm4^{C3/+}$; C3/C3: $Mcm4^{C3/C3}$; C3/C3 M2/+: $Mcm4^{C3/C3} Mcm2^{Gt/+}$. The dots in **(B)**, (D) and (F) represent individual biological replicates from independent experiments. p-values were determined by one-way ANOVA with Tukey's HSD test. Error bars represent mean ± SEM.

that the genome is replicated only once per cell cycle by preventing origin licensing after S phase initiation, and Geminin (GMNN) is a key factor in this process [36,37]. We observed significantly reduced GMNN in $Mcm4^{C3/C3}$ and $Mcm4^{C3/C3}$ $Mcm2^{Gt/+}$ TSCs compared to the WT (Fig 5E-F), which may facilitate their endoreduplication and premature differentiation into TGCs.

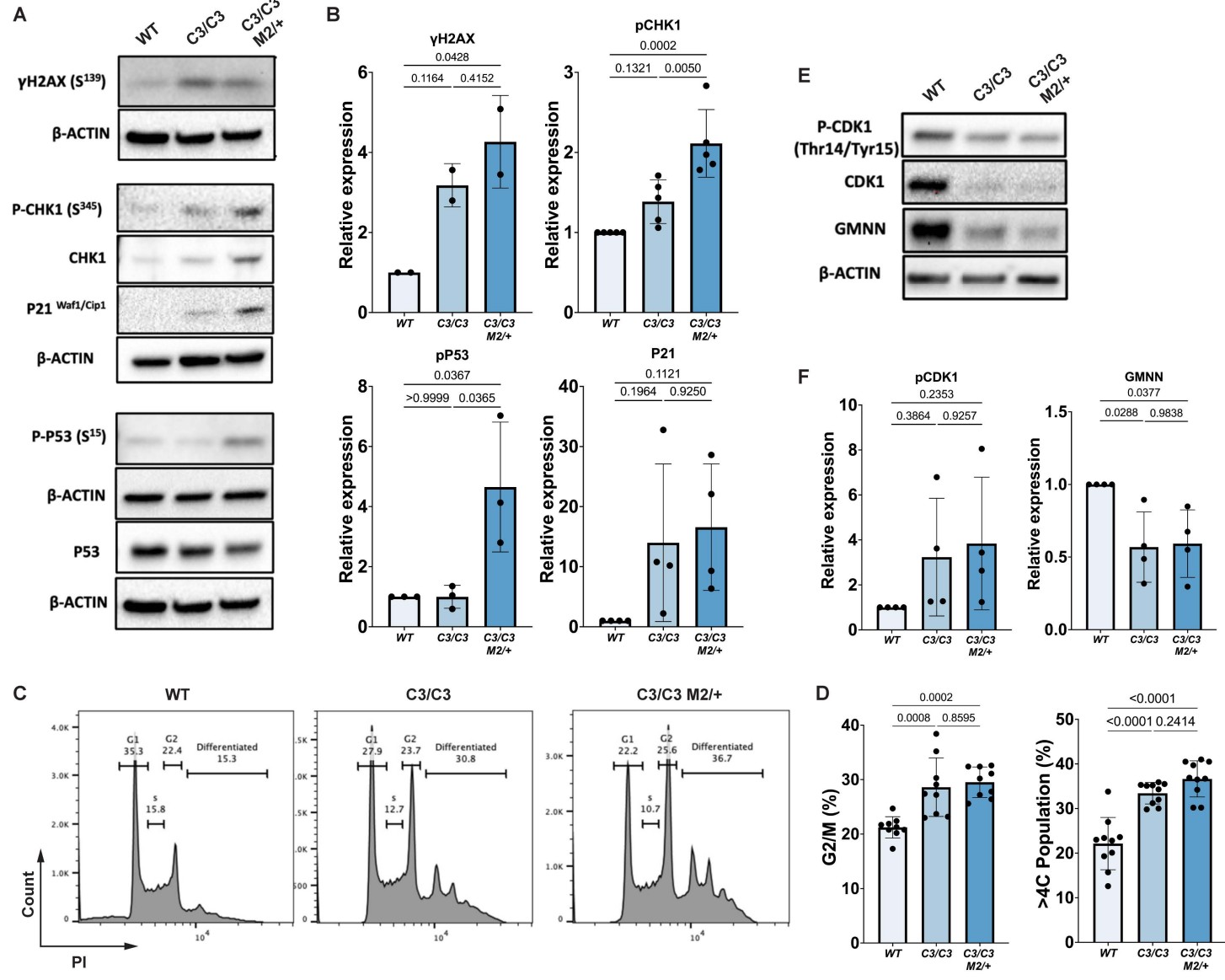

**Fig 5. Semi-lethal genotype TSCs activate DNA damage responses and prematurely differentiate. (A)** Western blot analysis of γH2AX, p-CHK1, CHK1, p21, p-p51(Ser15), and total p53 levels in TSCs of the indicated genotypes. **(B)** Relative expression of indicated proteins in different genotypes compared to controls. **(C)** Flow cytometry analysis of DNA content in TSCs. The >4C (polyploid) cell populations are labeled as "differentiated." **(D)** Quantification of DNA content and differentiated (>4C) cell populations across genotypes. **(E)** Western blot analysis of G2/M regulators in TSCs of the indicated genotypes. **(F)** Relative expression of indicated proteins in different genotypes compared to controls. *C3/C3*: $Mcm4^{C3/C3}$; *C3/C3 M2/+*: $Mcm4^{C3/C3}\ Mcm2^{Gt/+}$. The dots in (B) and (F) represent individual biological replicates. The dots in (D) represent independent experiments from different cell lines. p-values were determined by one-way ANOVA with Tukey's HSD test. Error bars represent mean±SEM.

## Decreasing MCM nuclear export factor MCM3 genetically rescues placental defects

*In vivo*, reducing MCM3 levels via heterozygosity paradoxically improved phenotypes associated with *Chaos3* mutants [17,21,22]. In addition to a reduction in cancer occurrence, *Mcm3* heterozygosity doubled the percentage of viable $Mcm4^{C3/C3}\ Mcm2^{Gt/+}$ animals at birth [21,22]. MCM3, which is an MCM nuclear export factor in yeast [38], increases

available MCMs for chromatin loading, thereby ameliorating RS [22]. Reduced MCMs in mice have been associated with deficits in stem cell compartments [39,40], and increasing the availability of chromatin-bound MCMs have been shown to improve the reprogramming efficiency of mouse embryonic fibroblasts [21]. Therefore, we reasoned that if the loss in cellularity in the JZ in MCM-deficient $Mcm4^{C3/C3}$ $Mcm2^{Gt/+}$ placentae was indeed caused by defective TSC maintenance, then reducing MCM3 in the $Mcm4^{C3/C3}$ $Mcm2^{Gt/+}$ background might partially rescue placenta phenotypes. Indeed, the placenta and embryo weights of E13.5 $Mcm4^{C3/C3}$ $Mcm2^{Gt/+}$ $Mcm3^{Gt/+}$ animals were significantly improved $vs.$ $Mcm4^{C3/C3}$ $Mcm2^{Gt/+}$ animals, rendering them similar to $Mcm4^{C3/C3}$ littermates (Fig 6A). The JZ area in $Mcm4^{C3/C3}$ $Mcm2^{Gt/+}$ $Mcm3^{Gt/+}$ placentae also significantly increased to the level of $Mcm4^{C3/C3}$ littermates (Fig 6B-C). These data suggest that the placental abnormalities were exacerbated by reduced MCMs, which is known to cause elevated GIN from insufficient numbers of dormant ("backup") origins [19,20,41].

## Discussion

Proper placentation is a prerequisite for embryo survival and a healthy pregnancy. Defects in the development of trophoblast cell types have been linked to pregnancy complications endangering both the mother and the fetus [42,43]. Here, we show that GIN caused by either MCM or FANCM deficiencies impairs JZ development (Figs 2A-D and S2). The labyrinth histology in GIN mutants exhibited reduced complexity, indicative of poor branching morphogenesis and/or underdeveloped blood spaces (Fig 2E-G). However, the impact of MCMs deficiency is more severe in the JZ than the labyrinth as the percentage of proliferating cells in the LZ were not affected (Fig 3A-B). The JZ cells act as hormonal and energy reservoirs for normal fetal development, and a smaller JZ is associated with intrauterine growth restriction (IUGR) [43]. Indeed, semilethal genotype embryos were significantly smaller compared to control littermates at mid-gestation (Fig 1C-H), and those that survived to birth also suffered from growth retardation [22].

Mutations decreasing MCM levels have been associated with reduced stem cell fitness [40,44], and our studies indicate that normal levels and biochemical functions of MCMs in mouse TSCs are crucial for proper placentation and embryogenesis. In mice, almost all JZ trophoblasts originate from cells in the ectoplacental cone, which arises from continued expansion of the extraembryonic ectoderm between E5.5 and E7.5 [42]. Polar trophectoderm in the developing blastocyst and cells in the extraembryonic ectoderm are considered to be the stem progenitors of all trophoblast cells that contribute to the mature placenta [42]. Under persistent RS, the percentage of proliferating cells in the EPC and ExE were significantly reduced at E7.5 (S5D-E Fig). TSC lines with high levels of GIN were derived inefficiently, and under conventional culture conditions, difficult to maintain, likely because such conditions do not adequately suppress differentiation signals [31]. Growth factors and inhibitors such as WNT inhibitor, ROCK inhibitor, and Activin A are often required to block differentiation pathways and promote self-renewal [32,33]. Use of such conditions enabled us to derive $Mcm4^{C3/C3}$ $Mcm2^{Gt/+}$ mutant TSCs, although they exhibited reduced proliferation potential and increased cell death (Fig 4A-D). Additionally, the observation of increased cells with >4C DNA content in mutant TSC cultures is consistent with premature differentiation into TGCs (Fig 5C-D). In contrast, ESCs bearing the same GIN genotypes did not exhibit reduced proliferation, poor stem cell maintenance or signs of DNA damage and cell death (S7 Fig). High levels of licensed dormant origins, fast origin licensing due to a short G1 phase, and ESCs specific replication-coupled mechanisms [34,45,46] likely explain why MCMs deficient ESCs show minimal cellular defects compared with mTSCs.

We propose that TSC defects in GIN genotypes were due to CDK1 inhibition resulting from activation of the CHK1-p53-p21 checkpoint pathway (Fig 5). Inhibition of CDK1, a key regulator of the G2/M transition, is sufficient to drive endoreduplication, promoting TSCs to differentiate into TGCs [9,35]. Consistently, we observed reduced CDK1 activity in TSCs with high levels of GIN (Fig 5F). On the other hand, TSCs were able to differentiate into JZ cells, namely SpTs and TGCs when prompted to differentiate (S6B Fig, TPBPA and PL1 immunoblots respectively). Taken together, we propose that due to poor TSC maintenance in semi-lethal genotypes, there was an insufficient pool of EPC cells that ultimately contributed to the poor cellularization of the JZ in the mature placenta. Consistent with the importance of robust MCM origin licensing

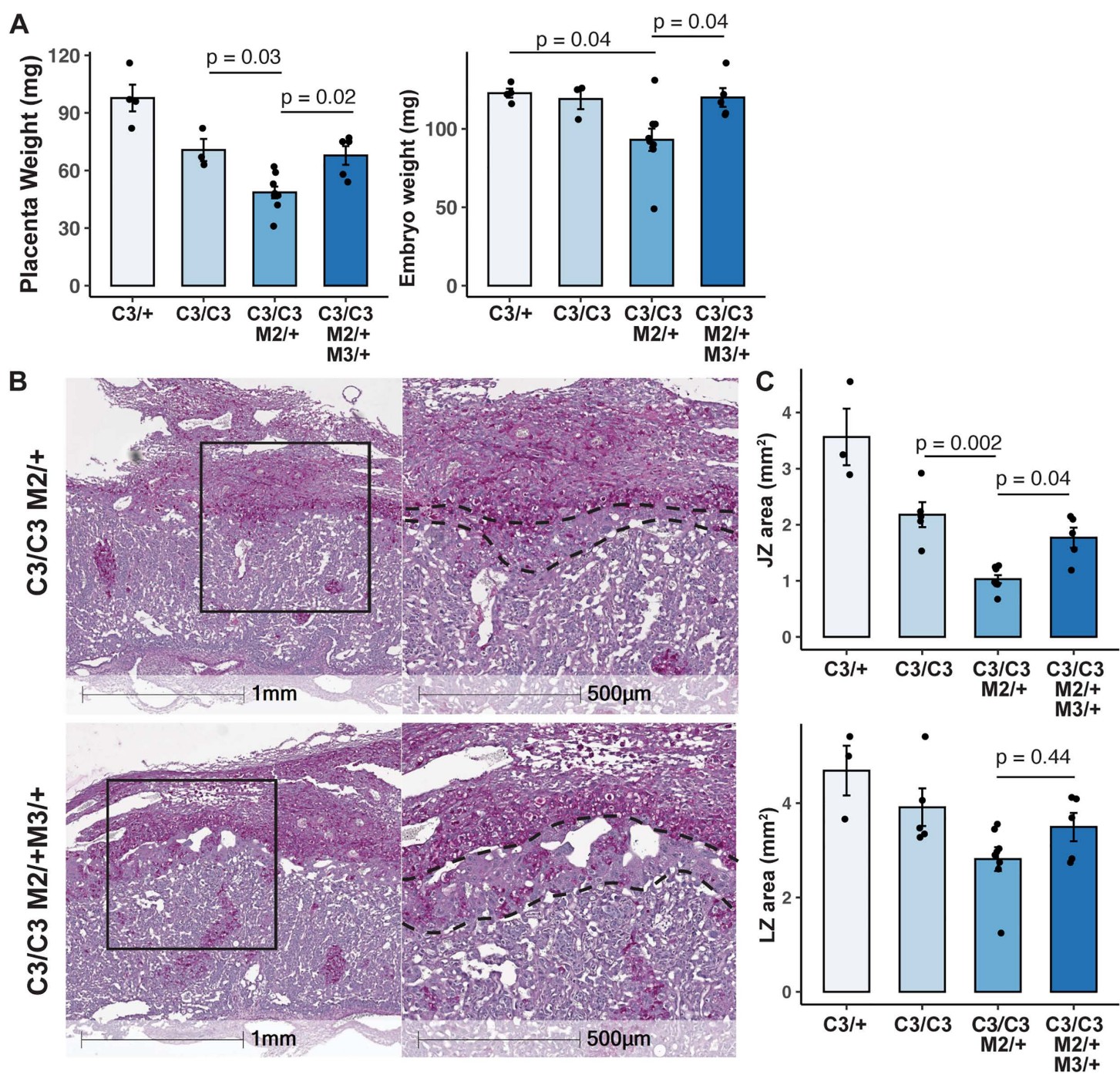

**Fig 6. *Mcm3* heterozygosity rescues placental defects in semi-lethal genotypes. (A)** Plots of placental and embryonic weights. Each data point is one placenta or an embryo. **(B)** PAS staining of E13.5 placental sections. **(C)** Average sizes of JZ and LZ areas. Each data point represents the average measurement taken from at least three sections of the same placenta. C3/+: $Mcm4^{C3/+}$; C3/C3: $Mcm4^{C3/C3}$; C3/C3 M2/+: $Mcm4^{C3/C3}Mcm2^{Gt/+}$; C3/C3 M2/+M3/+: $Mcm4^{C3/C3}Mcm2^{Gt/+} Mcm3^{Gt/+}$. *p-values* were determined by one-way ANOVA with Tukey's HSD test. ns: not significant. Error bar: SEM.

for stem cell maintenance, the placental size and the JZ defects were partially rescued by breeding mice that genetically rescue viability, presumably by increasing chromatin bound MCMs as shown in mutant MEFs (Fig 6) [21,22].

We were unable to rescue the viability of semilethal genotype embryos by deleting cytosolic nucleotide sensing pathways (S1 Table). Interestingly, the female-biased expression of genes involved in the inflammatory pathways and innate immune responses was abolished in STING-deficient semi-lethal placentae at E13.5 compared to wild type. Instead, we noted an overall increase in the inflammatory signatures in both male and female placentae from the STING-deficient semi-lethal genotypes (S8A Fig). This raises the possibility that a feedback mechanism might exist to regulate innate immune response in the placenta. We showed previously that treating pregnant dams with ibuprofen abolished sex bias and the viability of both male and female *Mcm4*<sup>C3/C3</sup> *Mcm2*<sup>Gt/+</sup> embryos was partially improved [17]. However, since ibuprofen treatment had very little noticeable impact on placental structure, we surmise that the action of ibuprofen provides symptomatic relief from the structural anomalies. Indeed, reduction in the JZ was more severe in the female semi-lethal genotype placentae than the males. Furthermore, genes predominantly expressed in the JZ cells were significantly underrepresented in the female placentae with semi-lethal genotypes (Figs 3C-D and S4D). Although the exact underlying mechanism of such sexual dimorphism needs further investigation, maternal factors, such as the epigenetic information carried by the oocyte, might account for sex-specific and parent-of-origin dependent placental defects [47]. Nevertheless, further experiments are needed to determine whether placental defects are the main cause of female biased embryonic semi-lethality in the *Chaos3* model.

MCMs are absolutely essential for DNA replication and the maintenance of genomic stability. While null alleles of any of the 6 MCMs cause pre-implantation lethality, hypomorphic alleles - such as the *Chaos3* allele of *Mcm4* - have enabled identification of tissue-specific sensitivies and impacts of compromised DNA replication [17,40,44]. Death of *Mcm4*<sup>C3/C3</sup> *Mcm2*<sup>Gt/+</sup> embryos are first detected after E9.5, a period in mouse development where embryonic lethality is typically associated with placental defects [48]. The Mouse Genome Informatics (MGI) database lists 106 genes (out of ~ 900) involved in genome maintenance whose deficiency causes embryonic lethality between E9.5-E14.5. Among these, only 26 of them were reported to have placentation defects, and the underlying molecular mechanism needs investigation (S3 Table). Furthermore, this is likely an underestimate as 70% of mouse knockout alleles that are embryonic lethal around this stage were reported to have abnormal placentation [48]. Taken together, we propose a modification to the conventional belief that trophoblast cells are resistant to GIN, instead genome maintenance mechanisms are likely at play at certain stages of placenta development in a lineage dependent manner. Whether RS compromises human placentation needs exploration.

## Materials and methods

### Ethics statement

Mouse studies in this work were approved by Cornell University's Institutional Animal Care and Use Committee under protocol number 2004–0038.

### Animal husbandry

All husbandry and mating experiments are conducted in the same animal facility and room, and under the same environmental conditions. *Tmem173*<sup>Gt</sup> [49] (JAX stock #017537) and *Myd88*- [50] (JAX stock #009088) mice were purchased from The Jackson Laboratory. *Ddx58* mutant mice were generated by CRISPR/Cas9-mediated genome editing as previously described [51]. Briefly, Cas9 protein (IDT Cat#: 1081058) and sgRNA (oligos containing both the crRNA: 5'-GATATCATT TGGATCAACTG-3' and the tracrRNA was purchased from IDT) were electroporated into single celled zygotes obtained from matings of wild-type C3HeB/FeJ mice (JAX stock #000658). The mice were maintained coisogenically in this strain background. For timed matings, female (8–12 weeks old) and male (8–16 weeks old) mice were placed in mating cages one day prior to checking for vaginal plugs. The presence of vaginal plugs was designated as E0.5. Genotyping of tissue biopsies was performed by Transnetyx (Memphis, TN, USA) or as previously published [22] using primers listed in S4 Table.

 **Genetics**

## Histology

Placental samples were isolated from E13.5 embryos and cut in half at midline. Half of a placenta was snap frozen in liquid nitrogen for RNA extraction, while the other half was fixed overnight in 4% paraformaldehyde in 1X PBS at 4$^0$C, dehydrated in 70% ethanol and embedded in paraffin blocks. Embedded samples were sectioned at 6μm thickness, deparaffinized in xylene, rehydrated in serial dilutions of ethanol solutions and finally in deionized water.

For Periodic Acid Schiff staining, rehydrated placental sections were incubated with 0.5% Periodic Acid for 5 minutes followed by 15 minutes incubation with Schiff's reagent (PAS staining kit, Sigma-Aldrich, Cat#: 1.01646). Sections were dipped into Hematoxylin Gill II (Leica Biosystems) for 20 seconds followed by 5 minutes in tap water. Slides were scanned and imaged on a Leica Aperio CS2 scanner. For measuring the junctional and labyrinth zone areas, at least six consecutive placental sections from the midline were stained, and the area of every other section was measured using FiJi/ImageJ software, and the average area was presented for each placenta.

For immunohistochemistry, deparaffinized and rehydrated sections were incubated with 3% hydrogen peroxide to quench endogenous peroxidase, then permeabilized with 0.1% Triton-X100 in 1X PBS at room temperature for 10 minutes, blocked for one hour with 3% goat serum in 1X PBS at room temperature. Incubation with primary antibodies was carried out overnight at 4$^0$C. Rabbit primary antibodies were detected with SignalStain Boost IHC detection reagent (CST Cat#: 8114). Sections were dipped into Hematoxylin Gill II (Leica Biosystems) for 20 seconds followed by 5 minutes in tap water. Slides were scanned using Leica Aperio CS2 scanner. Ki67 positive cells were quantified using QuPath (https://qupath.github.io/). All antibodies used in this study are listed in S5 Table.

## TSC derivation and culture

TSCs were derived from preimplantation E3.5 blastocysts. For the defined culture condition, individual E3.5 blastocysts were placed in 48-well plates containing a mouse embryonic fibroblast (MEF) feeder layer and cultured in the defined TS-stem medium (see below). By day 4, proliferating blastocyst outgrowths were observed. Between days 5 and 8 post-plating, these outgrowths were dissociated into single cells using Trypsin-EDTA. Initial TSC colonies appeared within four days after disaggregation. The feeder layers were kept during derivation, but were not used when growing established TSCs. The defined TS-stem medium (500 mL) was composed of 240 mL Neurobasal (Thermo Fisher; 21103049), 240 mL DMEM/F12 Ham's 1:1 (Thermo Fisher; 11320033), 2.5 mL N2 supplement 100X (Gibco; 17502048), 5 mL B-27 supplement 50X (Gibco; 17504044), 5 mL KnockOut serum replacement (Gibco; 10828028), 2.5 mL penicillin-streptomycin (Thermo Fisher; 15140122), 2.5 mL GlutaMAX Supplement (Gibco; 35050061), 30% bovine serum albumin (Tribioscience; TBS8031, 0.05% final concentration), $1.5 \times 10^{-4}$ M 1-thioglycerol (Sigma-Aldrich; 88640), 25–50 ng/mL recombinant mouse basic FGF (Peprotech; 450-33-50UG), 20 ng/mL Human/Mouse/Rat Activin A Recombinant Protein (Peprotech; 120-14E-100UG), 10 μM XAV939 (Selleckchem; S1180), and 5 μM Y27632 (Stemcell; 72304). The medium was refreshed every two days. For feeder-free culture, TSCs were plated on 0.2% Matrigel-coated dishes and incubated at 37°C for at least 1.5 hours or overnight before further culture.

For differentiation, TSCs were cultured in RPMI 1640 (Gibco; 11875085), 50 μg/mL penicillin-streptomycin (Gibco; 15140122), 2 mM L-glutamine (Gibco; 35050061), 1 mM sodium pyruvate (Gibco; 11360070), 100 μM β-mercaptoethanol (Sigma; M752), and 20% fetal bovine serum (Cytiva HyClone; SH30071.03HI).

## ESC derivation and culture

ESCs were derived as previously described [52]. Briefly, E3.5 blastocysts were seeded into individual wells of the 48-well plates containing MEF feeder layers. Between day 7–10, the blastocyst outgrowths were mechanically dissociated and seeded onto a new feeder plate. For regular maintenance of ESCs, the following ESCs media was used: DMEM (Corning,

10–017-CM) supplemented with 15% fetal bovine serum (Cytiva HyClone; SH30071.03HI), 1% penicillin-streptomycin (Gibco; 15140122), 1% sodium pyruvate (Gibco; 11360070), 1% MEM NEAA (Gibco, 11140050), 100 µM β-mercaptoethanol (Sigma; M752), 3 µM CHIR99021 (Tocris, 4423), 1 µM PD0325901, $10^3$ IU LIF (Peprotech, 250-02-25UG). For the initial establishment of the ESCs lines, the fetal bovine serum in ESCs media was replaced with the same percentage of KnockOut serum replacement (Gibco; 10828028).

### Immunofluorescence of cultured cells

Cells were grown on in chamber coverslips (iBidi µ-Slide 8 Well high; 80806) for 48 hours at 37°C with 5% $CO_2$, fixed in 4% paraformaldehyde for 15 min at room temperature, permeabilized with 0.1% Triton X-100 in 1X PBS for 15 min at room temperature, and blocked with 5% goat serum for one hour at room temperature. Cells were then incubated with the primary antibodies diluted in 0.1% BSA in 1X PBS overnight at $4^0$C, followed by the secondary antibody at room temperature for one hour. Nuclei were counterstained with DAPI diluted in 1X PBS. Slides were visualized using a Zeiss LSM 710 Confocal Microscope. Primary and secondary antibodies are listed in S5 Table.

### Western blotting

TSCs were trypsinized then washed twice with ice-cold 1×PBS before lysis in RIPA buffer (Thermo Fisher; 89900) supplemented with Complete Mini Protease Inhibitor Cocktail (Roche; 11836170001) and PhosSTOP Phosphatase Inhibitor Cocktail (Roche; 4906845001). TSC samples were sonicated three times at 25% amplitude for 10 seconds each (Thermo Fisher) and centrifuged at 14,000×g for 15 minutes at 4°C. The supernatant was collected, proteins were denatured at 95°C for 5 minutes in sample buffer, then run on Bio-Rad 4–20% Mini-PROTEAN Precast Gels (BioRad, Cat#: 4561094) at 100V for 75 minutes, then transferred onto PVDF membranes (Millipore; IPVH00010) on ice at 90V for 90 mins. Immunoblotting was performed using primary and secondary antibodies listed in S5 Table. Signal detection was carried out using enhanced chemiluminescence (Pierce; 32109) and visualized using a Bio-Rad Gel Doc XR Imaging System and analyzed with Image Lab (BioRad) software.

### Analysis of DNA content and cell cycle distribution

TSCs and differentiated trophoblast cells were harvested by trypsinization, washed twice with cold PBS, and fixed in 70% ethanol at -20°C for a minimum of two hours to preserve cellular DNA content. Fixed cells were then resuspended in 500 µL of propidium iodide (PI) staining solution, consisting of 0.1% Triton X-100, 1 µg/mL PI (Invitrogen; P3566), and 100 µg/mL RNase A (Thermo Fisher; EN0531) in 1×PBS. Following incubation at room temperature for 30 minutes in the dark, samples (100,000 events per sample) were analyzed on a BD FACSymphony A3 Cell Analyzer. Cell cycle distribution was determined using FlowJo software.

### EdU pulse labeling

For labeling TSCs, cells were treated with 10 µM EdU (5-ethynyl-2'-deoxyuridine) for 2 hours, and ESCs were treated for 30 minutes. The cells were then fixed in 4% paraformaldehyde for 15 minutes at room temperature, permeabilized with 0.1% Triton X-100 in PBS for 15 minutes, and stained using the Click-iT EdU Alexa Fluor 488 Imaging Kit (Thermo Fisher Scientific, C10337). Antibody labeling of cells was carried out after EdU detection. Nuclei were counterstained with DAPI to visualize total cell numbers. For TSCs, EdU and antibody labeled cells on the slides were imaged on Zeiss Axio Imager epifluorescence microscope. For ESCs, cells were analyzed on a BD FACSymphony A3 Cell Analyzer. The proportion of EdU-positive TSCs was quantified using FIJI/ImageJ software. The proportion of EdU-positive ESCs were analyzed using FlowJo software.

## Bulk RNA-seq and data analysis

Total RNA was isolated from WT and $Mcm4^{C3/C3}$ $Mcm2^{Gt/+}$ placentae at E13.5 using Zymo Quick-RNA mini-prep Kit (Zymo Research, Cat#: R1054). RNA sample quality was confirmed by spectrophotometry (Nanodrop) to determine concentration and chemical purity (A260/230 and A260/280 ratios) and with a Fragment Analyzer (Agilent) to determine RNA integrity. PolyA+RNA was isolated with the NEBNext Poly(A) mRNA Magnetic Isolation Module (New England Biolabs). UDI-barcoded RNA-seq libraries were generated with the NEBNext Ultra II Directional RNA Library Prep Kit (New England Biolabs). Each library was quantified with a Qubit (dsDNA HS kit; Thermo Fisher) and the size distribution was determined with a Fragment Analyzer (Agilent) prior to pooling. Libraries were sequenced on an Illumina NovaSeq6000 (2x150 PE reads). At least 20 million 150 bp paired-end reads were generated per library. For analysis, reads were trimmed for low quality and adaptor sequences with TrimGalore v0.6.0, a wrapper for Cutadapt v3.4 and FastQC v0.11.9 using parameters: -j 1 -e 0.1 --nextseq-trim=20 -O 1 -a AGATCGGAAGAGC --length 50 --fastqc. Unwanted reads were removed with STAR v2.7.0e using parameters: --outReadsUnmapped Fastx and final reads were mapped to the mouse reference genome Ensembl GRCm38 using STAR v2.7.0e. Read counts were then filtered to remove low abundance genes using iDEP 2.0 [53] default parameters. Differential expression analysis was performed using DESeq2 in iDEP 2.0 [53]. Genes were considered differentially expressed if they showed adjusted p-values (FDR) < 0.05 and absolute log2 fold change >1.5. Pairwise comparisons were conducted between wild type and mutant samples of the same sex. Gene set enrichment analyses were carried out in iDEP 2.0 [53]. RNA-seq data has been deposited to the Gene Expression Omnibus, accession GSE290784.

## Bulk RNA-Seq deconvolution

Deconvolution of bulk RNA-seq data was performed using BayesPrism [26] and publicly available placental scRNA-Seq data as a reference. The E13.5 placental single cell reference was generated using a publicly available raw count matrix. Following initial QC, doublet removal, and normalization, trophoblast and non-trophoblast cell types were identified through dimensionality reduction and clustering using Seurat V5 [54]. Cell types were annotated based on marker genes obtained from FindAllMarkers function (min.pct=0.05, min.diff.pct=0.05, logfc.threshold=0.15, return.thresh=0.1) together with reported cell type markers [24,25]. Mitochondrial and ribosomal genes were excluded from both the single cell and bulk RNA-Seq datasets. Deconvolution of the bulk data was performed by constructing a BayesPrism object with the *new.prism* function using filtered single-cell matrix as "reference" and our bulk RNA-Seq raw count matrix as "mixture". The posterior mean of cell type fraction theta was extracted using the get.fraction function and was plotted for comparison.

## Statistics

Statistical tests were carried out using one-way analysis of variance followed by Tukey's post hoc test or student's t-test using either R or Graphpad Prism. Graphs were generated using R or Adobe Illustrator. For each cell type, BayesPrism calculated fractions were analyzed using a generalized linear model with glht function in the multcomp package (https://CRAN.R-project.org/package=multcomp).

## Supporting information

**S1 Fig. Embryos with genomic instability genotypes have reduced placental and embryonic weight.** (A-C) Placental and embryonic weight as well as placental-to-embryonic weight ratio of each genotype from the reciprocal mating at E13.5. Females (circles), males (triangles). (D-F) Comparison of placental and embryonic weight as well as placental-to-embryonic weight ratio of male and female $Mcm4^{C3/C3}$ $Mcm2^{Gt/+}$ genotype from sex-skewing and reciprocal matings. Boxes indicate the maternal genotype. C3/+: $Mcm4^{C3/+}$; C3/C3: $Mcm4^{C3/C3}$; C3/+M2/+: $Mcm4^{C3/+}$ $Mcm2^{Gt/+}$; C3/C3 M2/+: $Mcm4^{C3/C3}$ $Mcm2^{Gt/+}$. p-values in (A-C) were calculated with one-way ANOVA followed by Tukey's HSD test.

p-values in (D-F) were calculated with two-way ANOVA. ns: not significant. *: $p < 0.05$; **: $p < 0.01$; ***: $p < 0.001$. Error bar: standard error of the mean. Each data point represents a single placenta or embryo.
(TIF)

**S2 Fig. *Fancm* deficiency results in placental defect.** (A-B) Placental and embryonic weights of indicated genotypes at E13.5. (C) Periodic Acid Schiff staining of E13.5 placental sections from indicated genotypes. (D) Measurements of placental JZ and LZ areas. p-values were calculated with one-way ANOVA followed by Tukey's HSD test. ns: not significant. *: $p < 0.05$; **: $p < 0.01$; ***: $p < 0.001$. Error bar: standard error of the mean. Each data point represents a single placenta or embryo.
(TIF)

**S3 Fig. Semi-lethal genotype placentae from reciprocal mating have a smaller JZ.** (A) Periodic Acid Schiff staining of placental sections from reciprocal matings at E13.5 for the indicated genotypes. (B-D) Measurements of placental JZ and LZ areas as well as the proportion of JZ from all indicated genotypes obtained from reciprocal matings at E13.5. (E-G) Comparison of JZ and LZ area as well as proportion of JZ in $Mcm4^{C3/C3}$ $Mcm2^{Gt/+}$ genotype from sex-skewing and reciprocal matings. Boxes above each graph indicate the maternal genotype. WT: wild type; C3/+: $Mcm4^{C3/+}$; C3/C3: $Mcm4^{C3/C3}$; C3/+M2/+: $Mcm4^{C3/+}$ $Mcm2^{Gt/+}$; C3/C3 M2/+: $Mcm4^{C3/C3}$ $Mcm2^{Gt/+}$. DC: decidua; JZ: junctional zone; LZ: labyrinth zone; p-TGC: parietal trophoblast giant cells; SpT: spongiotrophoblast; SynT: syncytiotrophoblast; GlyT: glycogen trophoblast; M: Male; F: Female. p-values were calculated with one-way ANOVA followed by Tukey's HSD test. ns: not significant. *: $p < 0.05$; **: $p < 0.01$; ***: $p < 0.001$. Error bar: standard error of the mean. Each data point in (B-G) represents the average measurement taken from at least three sections of the same placenta.
(TIF)

**S4 Fig. Deconvolution of placental bulk RNA-Seq.** (A) UMAP of E13.5 placental single-cell RNA-Seq. (B) Gene expression dot plot of marker genes for different cell types from reference placental scRNA-Seq. (C) Pairwise correlation of cell types based on the single cell transcriptome. (D) Cell type fractions identified by BayesPrism in control and semi-lethal genotype placentae. WT: wild type; C3/C3 M2/+: $Mcm4^{C3/C3}$ $Mcm2^{Gt/+}$. SpT: spongiotrophoblast; Gly: glycogen trophoblast; JZ_pre: junctional zone precursors; SynTI: syncytiotrophoblast I; SynTII: syncytiotrophoblast II; Lab_pre: labyrinth precursors; p-TGC: parietal trophoblast giant cells; S-TGC: sinusoidal trophoblast giant cells; Endo: endothelial cells; RBC: red blood cells. Each data point in (D) represents the cell type fraction calculated from one placental sample bulk RNA-Seq. Error bars in (D)represent standard error of the means, and each dot represents a single placental sample. *: $q < 0.05$ (Benjamini–Hochberg FDR adjusted q-value).
(TIF)

**S5 Fig. Placentae with replication stress genotypes have smaller junctional and labyrinth zones.** (A) Representative images of Cleaved-caspase 3 immunohistochemistry in E11.5 placental sections for the indicated genotypes. (B-C) Representative images of TPBPA and Cleaved-caspase 3 immunohistochemistry in E9.5 placental sections for the indicated genotypes. (D) Representative images of phospho-H3 immunofluorescence in E7.5 embryos sections. (E) Quantification of phospho-H3 positive cells in the epiblast and extraembryonic ectoderm. WT: wild type; C3/+: $Mcm4^{C3/+}$; C3/C3: $Mcm4^{C3/C3}$; C3/C3 M2/+: $Mcm4^{C3/C3}$ $Mcm2^{Gt/+}$. Epi: epiblast; ExE: extraembryonic ectoderm; EPC: ectoplacental cone; DC: decidua; LZ: labyrinth zone. Each data point in (E) represents the average percent positive cells from at least three consecutive sections of a single embryo.
(TIF)

**S6 Fig. Trophoblast stem cell derivation procedure in defined medium and premature differentiation of *Chaos3* mutant TSCs.** (A) Schematic representation of TSC derivation from E3.5 blastocysts under defined culture conditions. (B) Western blot analysis of trophoblast markers before and after TSC differentiation. Genotypes are indicated: C3, $Mcm4^{Chaos3/Chaos3}$; D, day; PL-1 and TPBPA are markers of TGCs and spongiotrophoblast, respectively. β-actin is a loading control.
(TIF)

**S7 Fig. Mouse embryonic stem cells (ESCs) are not sensitive to RS induced by MCM2–7deficiency.** (A) Flow cytometry analysis of naive pluripotency marker KLF4 in WT, C3/C3, and C3/C3 M2/＋ESCs. (B) Quantification of KLF4 positive cells in ESCs. (C) Flow cytometry analysis of EdU pulse labeled and KLF4 stained ESCs in indicated genotypes. (D)Quantification of the EdU and KLF4 double positive cells from flow cytometry. (E)Immunofluorescent analysis ofγH2AX in ESCs of the indicated genotypes. (F) Quantification of γH2Ax foci per nuclei in each indicated genotype. Each data point in (B), (D), (H) represents two technical replicates; dots in (F) represents foci per nuclei analyzed from individual images from two technical replicates. Error bar: standard error of the means. ns: not significant; p-values were calculated using one-way Anova.
(TIF)

**S8 Fig. STING deficiency in semi-lethal genotypes increases placental inflammation in both sexes.** (A) GSEA analysis of bulk RNA-Seq data from E13.5 placentas collected from *Mcm4$^{C3/C3}$ Mcm2$^{Gt/+}$* and *Mcm4$^{C3/C3}$ Mcm2$^{Gt/+}$ Tmem173$^{-/-}$* genotypes. Normalized enrichment scores of Hallmark inflammatory pathways in *Mcm4$^{C3/C3}$ Mcm2$^{Gt/+}$* or *Mcm4$^{C3/C3}$ Mcm2$^{Gt/+}$ Tmem173$^{-/-}$* compared to the wild type were plotted for both sexes. (B) Expression of genes involved in inflammatory pathways identified using leading edge analysis in male and female semi-lethal and STING-deficient semi-lethal genotype placentas compared to wild type, respectively.
(TIF)

**S1 Table. Numbers and crosses from indicated dam and sire genotypes.** Numbers in the topmost Table are from McNairn et al., *Nature,* (2019): specifically, offspring of dams homozygous for Chaos3 are taken from from "Table S3 Aggregate of crosses" from that publication, and offspring of heterozygous dams are from "Extended Data Table 1" from that same publication. *p-values* for the bottom 4 crosses were calculated using the $X^2$ probability test for the total number of pups. C3, Chaos3; Gt, gene trap allele of *Mcm2*.
(XLSX)

**S2 Table. TSC derivation efficiency under conventional and defined culture conditions.** The number of successfully derived WT, *Mcm4$^{C3/+}$*, and *Mcm4$^{C3/C3}$* TSC lines is shown for two independent experiments. Each condition contains 2 independent experiments. *p-values* were calculated using $X^2$ probability test.
(XLSX)

**S3 Table. Genome maintenance genes in MGI database with documented placental phenotypes when perturbed.**
(XLSX)

**S4 Table. Guide and Primer sequences.**
(XLSX)

**S5 Table. Antibody and cell labeling reagents.**
(XLSX)

**S1 Data. Primary data for Figures 1–6 and Supplemental Figures 1–5, 7, 8.**
(XLSX)

## Acknowledgments

We thank R. Munroe and C. Abratte of the Cornell Transgenic Core Facility for generating *Ddx58* knockout mice, and J. Grenier from Cornell's Genomics Facility at the Cornell Institute of Biotechnology for the RNA-seq experiments. We also thank Cornell's Imaging and Flow Cytometry cores.

## Author contributions

**Conceptualization:** John C. Schimenti.

**Formal analysis:** Mumingjiang Munisha, Rui Huang.

**Funding acquisition:** John C. Schimenti.

**Investigation:** Mumingjiang Munisha, Rui Huang, Jordan Khan.

**Methodology:** Rui Huang.

**Project administration:** John C. Schimenti.

**Writing – original draft:** Mumingjiang Munisha, Rui Huang.

**Writing – review & editing:** John C. Schimenti.

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
