## [Editor Report · Decision Letter 0]

6 Nov 2025

PGENETICS-D-25-01203

Chronic replication stress-mediated genomic instability disrupts placenta development in mice

PLOS Genetics

Dear Dr. Schimenti,

Thank you for submitting your manuscript to PLOS Genetics. After careful consideration, we feel that it has merit but does not fully meet PLOS Genetics's publication criteria as it currently stands. Therefore, we invite you to submit a revised version of the manuscript that addresses the points raised during the review process.

We look forward to receiving your revised manuscript.

Kind regards,

Fengwei Yu

Section Editor

PLOS Genetics

Fengwei Yu

Section Editor

PLOS Genetics

Aimée Dudley

Editor-in-Chief

PLOS Genetics

Anne Goriely

Editor-in-Chief

PLOS Genetics

**Journal Requirements:**

https://journals.plos.org/plosgenetics/s/submission-guidelines#loc-parts-of-a-submission

- TM on pages: 23, 24, and 25.

5) We have noticed that you have uploaded Supporting Information files, but you have not included a list of legends. Please add a full list of legends for your Supporting Information files after the references list.

Potential Copyright Issues:

i) Please confirm (a) that you are the photographer of 1B, or (b) provide written permission from the photographer to publish the photo(s) under our CC BY 4.0 license.

ii) Figures S6A, and S6B. Please confirm whether you drew the images / clip-art within the figure panels by hand. If you did not draw the images, please provide (a) a link to the source of the images or icons and their license / terms of use; or (b) written permission from the copyright holder to publish the images or icons under our CC BY 4.0 license. Alternatively, you may replace the images with open source alternatives. See these open source resources you may use to replace images / clip-art:

7) In the online submission form, you indicated that your data will be submitted to a repository upon acceptance. We strongly recommend all authors deposit their data before acceptance, as the process can be lengthy and hold up publication timelines. Please note that, though access restrictions are acceptable now, your entire minimal dataset will need to be made freely accessible if your manuscript is accepted for publication. This policy applies to all data except where public deposition would breach compliance with the protocol approved by your research ethics board. If you are unable to adhere to our open data policy, please kindly revise your statement to explain your reasoning and we will seek the editor's input on an exemption.

8) Please amend your detailed Financial Disclosure statement. This is published with the article. It must therefore be completed in full sentences and contain the exact wording you wish to be published.

2) If any authors received a salary from any of your funders, please state which authors and which funders..

**Reviewers' comments:**

**Figure resubmission:**
---

## [Decision Letter · Decision Letter 1]

4 Jan 2026

PGENETICS-D-25-01203R1

Chronic replication stress-mediated genomic instability disrupts placenta development in mice

PLOS Genetics

Dear Dr. Schimenti,

Thank you for submitting your manuscript to PLOS Genetics. After careful consideration, we feel that it has merit but does not fully meet PLOS Genetics's publication criteria as it currently stands. Therefore, we invite you to submit a revised version of the manuscript that addresses the points raised during the review process.

Please submit your revised manuscript within by Mar 05 2026 11:59PM. If you will need more time than this to complete your revisions, please reply to this message or contact the journal office at plosgenetics@plos.org. Please include the following items when submitting your revised manuscript:

We look forward to receiving your revised manuscript.

Kind regards,

Fengwei Yu

Section Editor

PLOS Genetics

Fengwei Yu

Section Editor

PLOS Genetics

Aimée Dudley

Editor-in-Chief

PLOS Genetics

Anne Goriely

Editor-in-Chief

PLOS Genetics

**Journal Requirements:**

1) Please amend your detailed Financial Disclosure statement. This is published with the article. It must therefore be completed in full sentences and contain the exact wording you wish to be published.

2) Thank you for stating 'The accession number for RNA-Seq data is GSE290784.'. We found that this accession number leads to a 'dataset not found' page. Kindly provide an active link to this data.

**Reviewers' comments:**

Reviewer's Responses to Questions

**Comments to the Authors:**

Reviewer #1: All of my concerns have been addressed.

Reviewer #2: General Comments

• On Page 10: “Analysis of publicly available single-cell RNA-Seq (scRNA-Seq) data from E13.5 mouse placenta revealed trophoblast and non-trophoblast cell types based on the most differentially expressed as well as previously known marker genes (S4A-B Fig).” This sentence does not contain information beyond what single cell analysis is, so it is not helpful to include. It is also unclear why S4A-B were included in this manuscript, as A and B are previously reported in the data source publication. If these are distinct, please include the details of the process of the annotations and how they are different from the source.

• Supplementary Fig 4C that includes the pairwise correlation of cell types by gene expression is unnecessary and does not add additional information relevant to this manuscript.

• Cell type deconvolution statistics between groups is done with incorrect statistical tests for comparing proportions. Please revise statistical test and restate conclusions on page 10. Also include pvalues for significant tests in the manuscript.

• Concerns remain for the bulk RNAseq analysis and statement of results. Given the difference in endothelial cell types between conditions, the bulk rnaseq analysis results are skewed by cell type. Please re-run the bulk RNAseq differential analysis including cell type proportions for those that are significant as covariates to remove the effect of those imbalances on the results. While running with covariates, it would also be helpful to run a model that includes both sexes and include sex and cell type proportion as covariates (e.g. ~0 + condition + endothelial_cell_proportion + sex). Limma-voom may be a more flexible tool for this model than DEseq2. Note: the inclusion of many covariates may be limited by sample size, so

• Report pvalues in the text for all of the statements that report a significant difference e.g. Page 15 “western blot analysis revealed elevated levels...”

Specific Comments

• Supp Figure 4D: include jitter in plots for individual samples. What does “mean fraction of cell types” represent? Shouldn’t there be a single fraction for each sample, so you are not reporting the mean?

• Page 27: sentence fragment “SARTools”

• Add software version numbers for Cutadapt, FASTQC, Fastx, iDEP, DESeq2

• Page 28 top – include a few more words for clarity of quantifying the bulk samples.

**Have all data underlying the figures and results presented in the manuscript been provided?**

Large-scale datasets should be made available via a public repository as described in the *PLOS Genetics*
data availability policy, and numerical data that underlies graphs or summary statistics should be provided in spreadsheet form as supporting information., and numerical data that underlies graphs or summary statistics should be provided in spreadsheet form as supporting information., and numerical data that underlies graphs or summary statistics should be provided in spreadsheet form as supporting information., and numerical data that underlies graphs or summary statistics should be provided in spreadsheet form as supporting information.

Reviewer #1: Yes

Reviewer #2: Yes

PLOS authors have the option to publish the peer review history of their article (what does this mean?). If published, this will include your full peer review and any attached files.). If published, this will include your full peer review and any attached files.). If published, this will include your full peer review and any attached files.). If published, this will include your full peer review and any attached files.

...

Reviewer #1: **Yes:** William PastorWilliam PastorWilliam PastorWilliam Pastor

Reviewer #2: No

**Figure resubmission:**
---

## [Decision Letter · Decision Letter 2]

31 Mar 2026

Dear Dr Schimenti,

We are pleased to inform you that your manuscript entitled "Chronic replication stress-mediated genomic instability disrupts placenta development in mice" has been editorially accepted for publication in PLOS Genetics. Congratulations!

Yours sincerely,

Fengwei Yu

Section Editor

PLOS Genetics

Fengwei Yu

Section Editor

PLOS Genetics

Aimée Dudley

Editor-in-Chief

PLOS Genetics

Anne Goriely

Editor-in-Chief

PLOS Genetics

BlueSky: @plos.bsky.social

Comments from the reviewers (if applicable):

Reviewer's Responses to Questions

**Comments to the Authors:**

Reviewer #2: The authors have addressed all reviewer comments.

**Have all data underlying the figures and results presented in the manuscript been provided?**

Large-scale datasets should be made available via a public repository as described in the *PLOS Genetics*
data availability policy, and numerical data that underlies graphs or summary statistics should be provided in spreadsheet form as supporting information., and numerical data that underlies graphs or summary statistics should be provided in spreadsheet form as supporting information., and numerical data that underlies graphs or summary statistics should be provided in spreadsheet form as supporting information., and numerical data that underlies graphs or summary statistics should be provided in spreadsheet form as supporting information.

Reviewer #2: Yes

PLOS authors have the option to publish the peer review history of their article (what does this mean?). If published, this will include your full peer review and any attached files.). If published, this will include your full peer review and any attached files.). If published, this will include your full peer review and any attached files.). If published, this will include your full peer review and any attached files.

...

Reviewer #2: No

**Data Deposition**

If you have submitted a Research Article or Front Matter that has associated data that are not suitable for deposition in a subject-specific public repository (such as GenBank or ArrayExpress), one way to make that data available is to deposit it in the Dryad Digital Repository. As you may recall, we ask all authors to agree to make data available; this is one way to achieve that. A full list of recommended repositories can be found on our . As you may recall, we ask all authors to agree to make data available; this is one way to achieve that. A full list of recommended repositories can be found on our . As you may recall, we ask all authors to agree to make data available; this is one way to achieve that. A full list of recommended repositories can be found on our . As you may recall, we ask all authors to agree to make data available; this is one way to achieve that. A full list of recommended repositories can be found on our website....

http://datadryad.org/submit?journalID=pgenetics&manu=PGENETICS-D-25-01203R2

Additionally, please be aware that our data availability policy requires that all numerical data underlying display items are included with the submission, and you will need to provide this before we can formally accept your manuscript, if not already present. requires that all numerical data underlying display items are included with the submission, and you will need to provide this before we can formally accept your manuscript, if not already present. requires that all numerical data underlying display items are included with the submission, and you will need to provide this before we can formally accept your manuscript, if not already present. requires that all numerical data underlying display items are included with the submission, and you will need to provide this before we can formally accept your manuscript, if not already present.

**Press Queries**

If you or your institution will be preparing press materials for this manuscript, or if you need to know your paper's publication date for media purposes, please inform the journal staff as soon as possible so that your submission can be scheduled accordingly. Your manuscript will remain under a strict press embargo until the publication date and time. This means an early version of your manuscript will not be published ahead of your final version. PLOS Genetics may also choose to issue a press release for your article. If there's anything the journal should know or you'd like more information, please get in touch via plosgenetics@plos.org....

---

## [Editor Report · Acceptance letter]

PGENETICS-D-25-01203R2

Chronic replication stress-mediated genomic instability disrupts placenta development in mice

Dear Dr Schimenti,

We are pleased to inform you that your manuscript entitled "Chronic replication stress-mediated genomic instability disrupts placenta development in mice" has been formally accepted for publication in PLOS Genetics! Your manuscript is now with our production department and you will be notified of the publication date in due course.

With kind regards,

Olena Szabo

PLOS Genetics

On behalf of:
